# A robust and tunable mitotic oscillator in artificial cells

Ye Guan[1,2†], Zhengda Li[1,3†], Shiyuan Wang[1], Patrick M Barnes[4], Xuwen Liu[5], Haotian Xu[6], Minjun Jin[7], Allen P Liu[1,8], Qiong Yang[1,3,4*]

[1]Department of Biophysics, University of Michigan, Ann Arbor, United States; [2]Department of Chemistry, University of Michigan, Ann Arbor, United States; [3]Department of Computational Medicine and Bioinformatics, University of Michigan, Ann Arbor, United States; [4]Department of Physics, University of Michigan, Ann Arbor, United States; [5]Department of Physics, University of Science and Technology of China, Hefei Shi, China; [6]Department of Computer Science, Wayne State University, Detroit, United States; [7]Department of Biological Chemistry, University of Michigan, Ann Arbor, United States; [8]Department of Mechanical Engineering, University of Michigan, Ann Arbor, United States

**Abstract** Single-cell analysis is pivotal to deciphering complex phenomena like heterogeneity, bistability, and asynchronous oscillations, where a population ensemble cannot represent individual behaviors. Bulk cell-free systems, despite having unique advantages of manipulation and characterization of biochemical networks, lack the essential single-cell information to understand a class of out-of-steady-state dynamics including cell cycles. Here, by encapsulating *Xenopus* egg extracts in water-in-oil microemulsions, we developed artificial cells that are adjustable in sizes and periods, sustain mitotic oscillations for over 30 cycles, and function in forms from the simplest cytoplasmic-only to the more complicated ones involving nuclear dynamics, mimicking real cells. Such innate flexibility and robustness make it key to studying clock properties like tunability and stochasticity. Our results also highlight energy as an important regulator of cell cycles. We demonstrate a simple, powerful, and likely generalizable strategy of integrating strengths of single-cell approaches into conventional in vitro systems to study complex clock functions.
DOI: https://doi.org/10.7554/eLife.33549.001

*For correspondence:
qiongy@umich.edu

†These authors contributed equally to this work

Competing interests: The authors declare that no competing interests exist.

## Introduction

Spontaneous progression of cell cycles represents one of the most extensively studied biological oscillations. Cytoplasmic extracts predominantly from *Xenopus* eggs (*Murray, 1991*) have made major contributions to the initial discovery and characterization of the central cell-cycle regulators including the protein complex cyclin B1-Cdk1 (*Masui and Markert, 1971*; *Murray et al., 1989*; *Lohka and Maller, 1985*; *Lohka et al., 1988*) and the anaphase-promoting complex or cyclosome (APC/C) (*Sudakin et al., 1995*). Cell-free extracts have also been used to investigate downstream mitotic events such as spindle assembly and chromosome segregation (*Hannak and Heald, 2006*). Moreover, detailed dissections of the regulatory circuits in these extracts have revealed architecture of interlinked positive and negative feedbacks (*Kumagai and Dunphy, 1992*; *Mueller et al., 1995*; *Yang and Ferrell, 2013*; *Chang and Ferrell, 2013*; *Trunnell et al., 2011*; *Kim and Ferrell, 2007*; *Pomerening et al., 2005*; *Pomerening et al., 2003*; *Novak and Tyson, 1993*; *Thron, 1996*) (*Figure 1A*). Such interlinked feedback loops are also found in many other biological oscillators (*Rust et al., 2007*; *Hoffmann et al., 2002*; *Cross, 2003*; *Lee et al., 2000*) and have been shown computationally to play an important role in achieving the essential clock functions such as

robustness and tunability (*Tsai et al., 2008*). These studies have stimulated major interests in quantitative characterization of clock functions, for which an experimental platform is still lacking.

Compared to in vivo systems, circuits reconstituted in cell-free extracts contain well-defined recombinant molecules and are more amenable to systematic design, manipulation and quantitative biochemical measurements. However, one major limitation for most in vitro reconstitutions up to date is that oscillations are generated in well-mixed bulk solutions, which tend to produce quickly damped oscillations (*Pomerening et al., 2005*; *Nakajima et al., 2005*). Additionally, these bulk reactions lack the similarity to the actual cell dimensions and the ability of mimicking spatial organization achieved by functional compartmentalization in real cells. These limitations make it impossible to retrieve the cellular heterogeneity to investigate important and challenging questions, such as stochasticity and tunability of an oscillator.

To overcome these challenges, we developed an artificial cell cycle system by encapsulating reaction mixtures containing cycling *Xenopus* egg cytoplasm (*Murray, 1991*) in cell-scale micro-emulsions. These droplet-based cells are stable for days and keep oscillating for dozens of cycles, offering large gains in high-throughput and long-term tracking of dynamical activities in individual droplets. In this system, we successfully reconstituted a series of mitotic events including chromosome condensation, nuclear envelope breakdown and destruction of anaphase substrates such as the proteins securin and cyclin B1. The oscillation profiles of the system such as period and number of cycles can be reliably regulated by the amount of cyclin B1 mRNAs or sizes of droplets. Additionally, we found that energy may be a critical factor for cell cycle behaviors.

## Results and discussion

### The oscillator reliably drives the periodic progression of multiple mitotic events

To create a cell-like in vitro mitotic system, we used a simple vortexing technique (*Ho et al., 2017*) to compartmentalize reactions of cycling *Xenopus* egg extracts (*Murray, 1991*) into microemulsion droplets, with radii ranging from 10 μm to 300 μm (*Figure 1B*, Materials and methods). The droplets were loaded into a Teflon-coated chamber and recorded using long-term time-lapse fluorescence microscopy. The fluorescence time courses of each droplet were analyzed to obtain information of period, amplitude, number of cycles, droplet size, etc.

To examine the functionality of the droplet mitotic system, we added de-membranated sperm chromatin, purified green fluorescent protein-nuclear localization signal (GFP-NLS), securin-mCherry mRNA and Hoechst 33342 dye to the cytoplasmic extracts. We demonstrated that the system is capable of reconstructing at least three mitotic processes in parallel that alternate between interphase and mitosis (*Figure 1—video 1*). An example artificial cell that undergoes these mitotic processes is shown in *Figure 1C*. The autonomous alternation of distinct cell-cycle phases is driven by a self-sustained oscillator, the activity of which was indicated by the periodic degradation of an anaphase substrate of APC/C, securin-mCherry. In interphase, the presence of sperm chromosomal DNA, labeled by Hoechst, initiated the self-assembly of a nucleus, upon which GFP-NLS protein was imported through the nuclear pores. The spatial distributions of Hoechst and GFP-NLS thus coincided in an interphase nucleus (*Figure 1C* columns 1, 3, and 5). As the artificial cell entered mitosis, the chromosome condensed resulting in a tighter distribution of Hoechst, while the nuclear envelope broke down and GFP-NLS quickly dispersed into a uniform distribution in the whole droplet (*Figure 1C* columns 2 and 4). The time courses for these processes were analyzed in *Figure 1D*, indicating that the chromosome condensation and nuclear envelop breakdown (NEB) happened almost at the same time, while securin degradation lagged behind these two processes at each cycle. All together, these experiments showed that the droplet system successfully reconstituted a cell-free mitotic oscillator centered on Cdk1 and APC/C that can reliably drive the periodic progression of downstream events including chromosome morphology change and nuclear envelope breakdown and re-assembly, like what occurs in vivo.

### The oscillator is effectively tunable in frequency with cyclin B1 mRNAs

The ability to adjust frequency is an important feature shared by many oscillators (*Tsai et al., 2008*). Here, we demonstrated that the present system provides an effective experimental solution to the

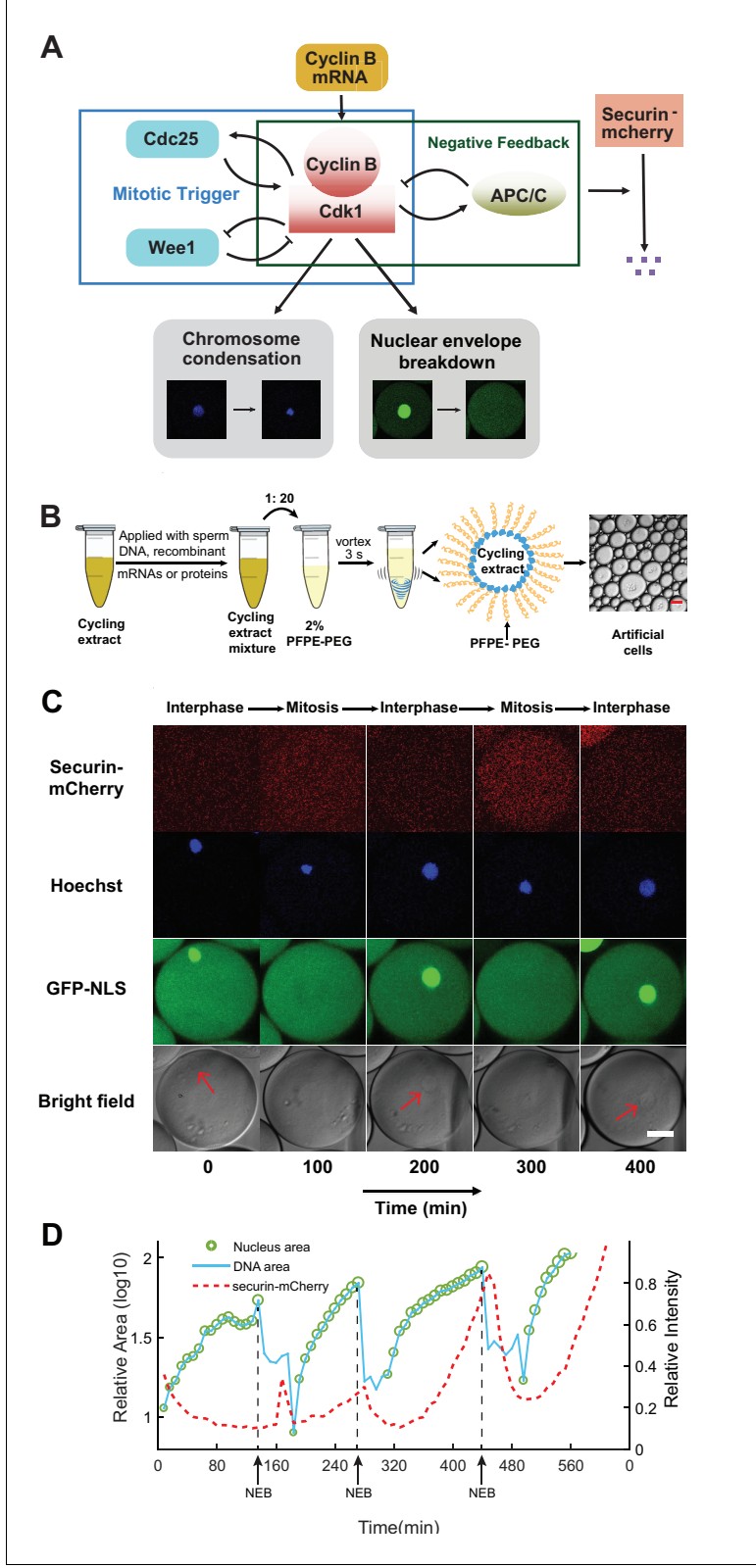

**Figure 1.** Reconstitution of an in vitro cell cycle clock and downstream mitotic events. (**A**) Schematic view of a cell cycle oscillator that consists of coupled positive and negative feedback loops. The central regulator, cyclin B-Cdk1 complex activates its own activator, phosphatase Cdc25, forming a positive feedback loop, and inhibits its own inhibitor, kinase Wee1, forming a double negative feedback loop. Additionally, cyclinB-Cdk1 activates the E3

*Figure 1 continued on next page*

*Figure 1 continued*

ubiquitin ligase APC/C, which targets cyclin B for degradation and completes a core negative feedback loop. Active APC/C also promotes the degradation of another substrate securin. Once cyclinB1-Cdk1 complex is activated, the circuit drives a set of mitotic events including chromosome condensation and nuclear envelope breakdown (NEB). (B) Experimental procedures. Cycling *Xenopus* extracts are supplemented with various combinations of recombinant proteins, mRNAs, and de-membraned sperm DNAs, which are encapsulated in 2% Perfluoropolyether-poly (ethylene glycol) (PFPE-PEG) oil microemulsions. Scale bar is 100 µm. (C) Snapshots of a droplet were taken periodically both in fluorescence channels (top three rows) and bright-field (the last row). The cyclic progression of the cell cycle clock and its downstream mitotic processes are simultaneously tracked by multiple fluorescence reporters. The clock regulator APC/C activity is reported by its substrate securin-mCherry, chromosomal morphology changes by the Hoechst stains, and NEB by GFP-NLS. Nuclear envelopes (red arrows) are also detectable on bright field images, matching the localization of GFP-NLS indicated nuclei. Scale bar is 30 µm. (D) Multi-channel measurements for the droplet in *Figure 1C*. The nucleus area (green circle) is calculated from the area of the nuclear envelope indicated by GFP-NLS, noting that the areas of the green circles are also scaled with the real areas calculated for the nuclei. DNA area curve (blue line) shows the chromosome area identified by Hoechst 33342 dye. Chromosome condensation happens almost at the same time as the nuclear envelope breaks down (black dashed line). The red dashed line represents the intensity of securin-mCherry over time, suggesting that degradation of the APC/C substrate lags behind NEB consistently at each cycle.

DOI: https://doi.org/10.7554/eLife.33549.002

The following video is available for figure 1:

**Figure 1—video 1.** Reconstitution of cell cycle clock and mitotic events.

DOI: https://doi.org/10.7554/eLife.33549.003

study of tunability of the clock. To avoid any interference from the complicated nuclear dynamics, we reconstituted a minimal mitotic oscillatory system which, in the absence of sperm chromatin, formed no nuclei. This simple, cytoplasmic-only oscillator produced highly robust, undamped, self-sustained oscillations up to 32 cycles over a lifetime of 4 days (*Figure 2A,B*, *Figure 2—video 1*), significantly better than many existing synthetic oscillators.

To modulate the speed of the oscillations, we supplied the system with various concentrations of purified mRNAs of full-length cyclin B1 fused to YFP (cyclin B1-YFP), which function both as a reporter of APC/C activity and as an activator of Cdk1. A droplet supplied with both cyclin B1-YFP and securin-mCherry mRNAs exhibited oscillations with highly correlated signals (*Figure 2C*, *Figure 2—video 2*), suggesting that both are reliable reporters for the oscillator activity. With an increased concentration of cyclin B1-YFP mRNAs added to the system, we observed a decrease in the average period (*Figure 2D*, *Figure 2—source data 1*), meaning that a higher cyclin B1 concentration tends to speed up the oscillations. However, the average number of cycles (*Figure 2E*, *Figure 2—source data 1*) was also reduced with increased cyclin B1 concentrations, resulting in a negative correlation between the lifetime of oscillations and the amount of cyclin B1 mRNAs. The extracts will eventually arrest at a mitotic phase in the presence of high concentrations of cyclin B1.

## The behavior of the single droplet oscillator is size-dependent

Moreover, this system provides high flexibility in analyzing droplets with radii ranging from a few µm to 300 µm, enabling characterization of size-dependent behaviors of cell cycles. At the scale of a cell, the dynamics of biochemical reactions may become stochastic. Although stochastic phenomenon has been studied extensively in genetic expressions, studying a stochastic system that is out of steady-state can be challenging in living organism due to low throughput and complications from cell growth, divisions and other complex cellular environments. These limitations can be overcome by reconstitution of in vitro oscillators inside cell-scale droplets, which are in absence of cell growth and divisions. Parallel tracking of droplets also enables high-throughput data generation for statistical analysis. In vitro compartmentalization of molecules, especially rate-limiting molecules such as cyclin B1 mRNAs, into cell-sized droplets may have a major effect on the reaction kinetics of cell cycles. The smaller the size of a droplet, the smaller the copy number of molecules encapsulated inside the droplet and the larger the inherent stochasticity of biochemical reactions. Additionally, the partition errors of these molecule resulted from compartmentalization may further contribute to the variations of droplet behaviors in a size-dependent manner. *Figure 2F* (*Figure 2—source data 2*) showed that smaller droplets led to slower oscillations with a larger variance of the periods,

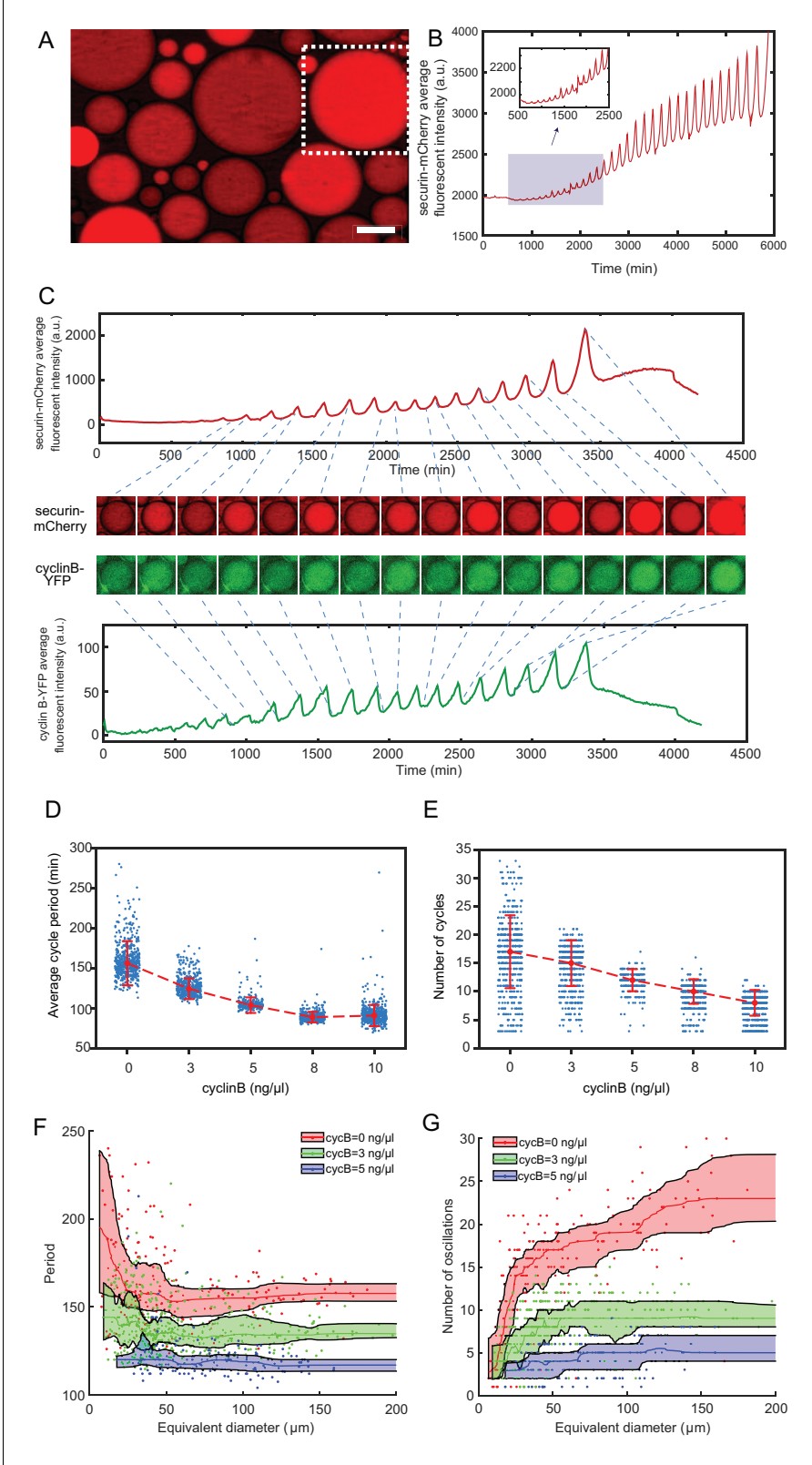

**Figure 2.** The minimal cell cycle oscillator is robust and tunable. (**A**) Fluorescence image of securin-mCherry, a reporter for the cell cycle oscillator, in micro-emulsion droplets (scale bar, 100 μm). One example droplet (inside the white dotted framed square) is selected for time course analysis in *Figure 2B*. (**B**) The time course of securin-mCherry fluorescence intensity of the selected droplet from *Figure 2A*, indicating 32 undamped oscillations over
*Figure 2 continued on next page*

*Figure 2 continued*

a course of 100 hr. (C) Simultaneous measurements of fluorescence intensities of securin-mCherry (upper panel) and cyclin B-YFP (lower panel), showing sustained oscillations for about 58 hr. The mRNA concentrations of securin-mCherry and cyclin B-YFP are 10 ng/μL and 1 ng/μL. The series of mCherry and YFP images correspond to selected peaks and troughs in the time courses of fluorescence intensities. The two channels have coincident peaks and troughs for all cycles, suggesting that they both are reliable reporters for the cell cycle oscillator. (D, E) The oscillator is tunable in frequency (D) and number of cycles (E) as a function of the concentration of cyclin B mRNAs. Cyclin B not only functions as a substrate of APC/C but also binds to Cdk1 for its activation, functioning as an 'input' of the clock. In *Figure 2D*, the cell cycle periods are shortened by increasing the mRNA concentrations. In *Figure 2E*, the number of total cell cycles is reduced in response to increasing cyclin B mRNA concentrations. Each data point represents a single droplet that was collected from one of the loading replicates (see Materials and methods 7). Red dashed line connects medians at different conditions. Error bar indicates median absolute deviation (MAD). (F, G) Droplets with smaller diameters have larger periods on average and a wider distribution of periods (F), and exhibit smaller number of oscillations on average (G). Colored areas represent moving 25 percentiles to 75 percentiles and is smoothed using the LOWESS smoothing method (see Materials and methods 7). The equivalent diameter is defined as the diameter of a sphere that has an equal volume to that of a droplet, estimated by a volume formula in literature (*Good et al., 2013*). Note that these size effects are smaller for droplets with higher cyclin B mRNA concentrations.

DOI: https://doi.org/10.7554/eLife.33549.004

The following video and source data are available for figure 2:

**Source data 1.** Source data for generating *Figure 2D-E*.
DOI: https://doi.org/10.7554/eLife.33549.005
**Source data 2.** Source data for generating *Figure 2F-G*.
DOI: https://doi.org/10.7554/eLife.33549.006
**Figure 2—video 1.** Free-running in vitro cell cycles detected by securin-mCherry reporter.
DOI: https://doi.org/10.7554/eLife.33549.007
**Figure 2—video 2.** Tuning of the clock speed.
DOI: https://doi.org/10.7554/eLife.33549.008

consistent with the size effect reported on an in vitro transcriptional oscillator (*Weitz et al., 2014*). We also observed a reduced number of oscillations and a smaller variance of the cycle number in smaller droplets (*Figure 2G*, *Figure 2—source data 2*). Interestingly, these size effects become less dramatic for droplets with larger sizes or with higher concentrations of cyclin B1 mRNAs.

## Energy depletion model recapitulates dynamics of the oscillator

The results in *Figure 2D–G* indicated that the system is tunable by cyclin B1 mRNA concentration and droplet size in different manners. Although the period and number of cycles responded to varying droplet sizes in opposite directions, they followed the same trend when modulated by cyclin B1 mRNAs, resulting in a lifespan of the oscillatory system sensitive to cyclin B1 mRNA concentration. Moreover, we have observed that securin-mCherry and cyclin B1-YFP both exhibited oscillations of increased amplitude, baseline, and period over time (*Figure 2C*), of which the increasing period over time is evident by analysis in *Figure 3—figure supplement 1A and B*. These trends cannot be explained by existing cell cycle models (*Yang and Ferrell, 2013*; *Tsai et al., 2014*).

Unlike intact embryos, cell-free extracts lack yolk as an energy source and lack sufficient mitochondria for energy regeneration. We postulated that energy is an important regulator for a droplet system with a limited amount of energy source consumed over time. To gain insights into our experimental observations and better understand the in vitro oscillator system, we built a model to examine how energy consumption plays a role in modulating the oscillation behaviors. The energy depletion model is based on a well-established cell-cycle model (*Yang and Ferrell, 2013*; *Tsai et al., 2014*) modified by introducing ATP into all phosphorylation reactions (*Figure 3A*, Materials and methods 8 and 9).

In the cell cycle network, the activation of Cdk1 is co-regulated by a double positive feedback through a phosphatase Cdc25 and a double negative feedback through a kinase Wee1. The balance between Wee1 and Cdc25 activity was suggested to be crucial for the transition of cell cycle status during early embryo development (*Tsai et al., 2014*). In light of this, we defined the balance between Wee1 and Cdc25 by the ratio $R = \frac{k_{Wee1}[Wee1]}{k_{Cdc25}[Cdc25]}$. We noted that ATP-dependent

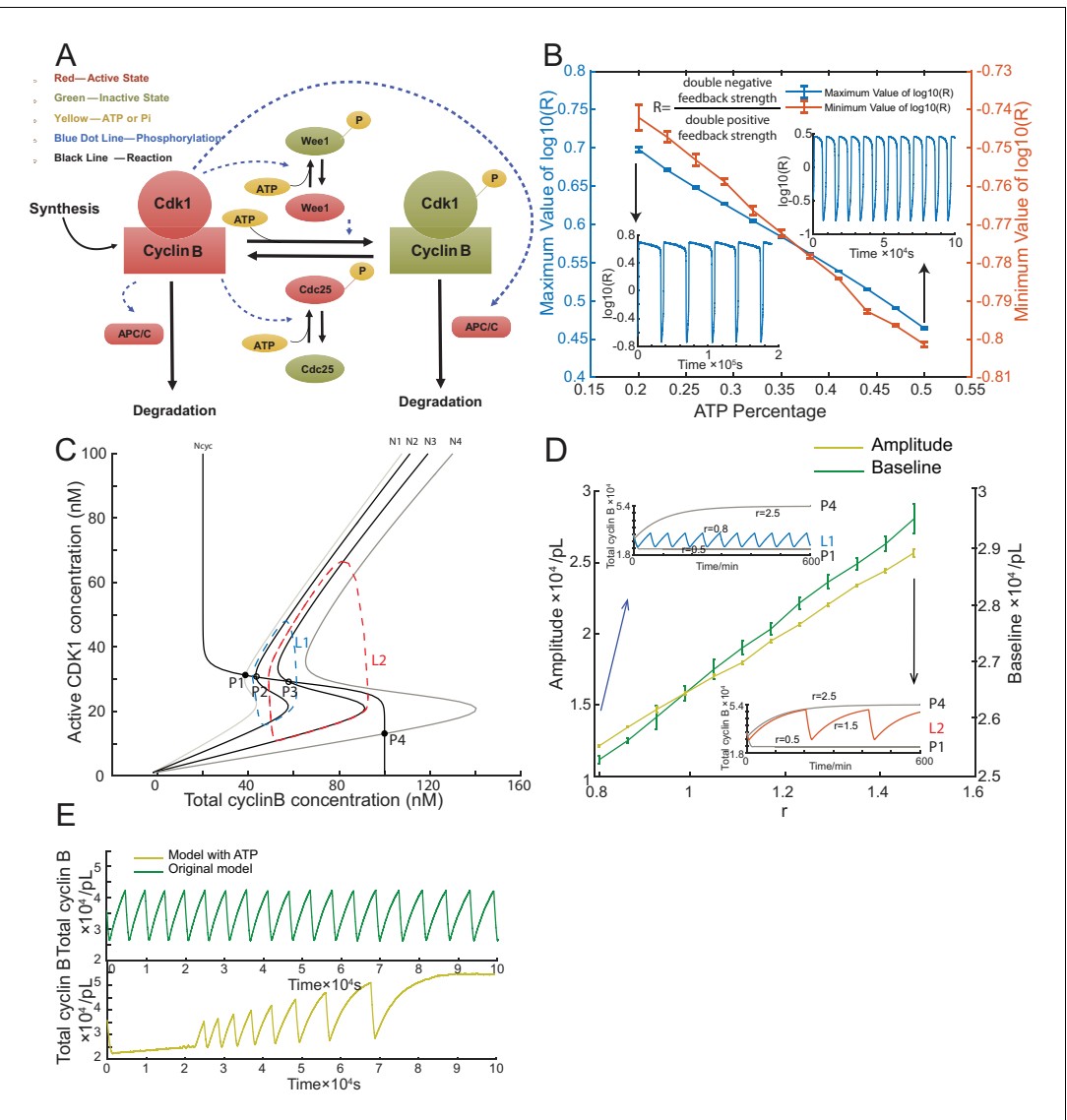

**Figure 3.** Simulation of cell cycle model with energy depletion. (**A**) Schematic view of the cyclin B-Cdk1 oscillation system. Note that ATP is taken into consideration. Activated molecules are marked in red, inactivated molecules in green and ATP or Pi in yellow. Black line indicates a reaction and blue dotted line a phosphorylation. (**B**) Relationship between ATP percentage and R value, showing that decreasing the ATP concentration leads to a higher R value. Error bars represent ranges from three simulations. Two inserts represent the dynamics of R value over time when the ATP percentage [ATP]/([ATP]+[ADP]) is set as 0.2 (left) and 0.5 (right). The model is simulated using Gillespie algorithm. (**C**) Phase plots of the two-ODE model. Parameters for the cyclin B nullcline (Ncyc) (**Yang and Ferrell, 2013**) and the Cdk1 nullclines with a variety of values of r (N1, r = 0.5; N2, r = 0.8; N3, r = 1.5; N4, r = 2.5) were chosen based on previous experimental work (**Pomerening et al., 2003**; **Sha et al., 2003**). Note that the r here is a parameter and is different from R in **Figure 3B**. Two sample traces of limit cycle oscillations are plotted for r = 0.8 (L1) and r = 1.5 (L2), showing that a larger r value leads to a higher amplitude and baseline. In addition, r = 0.5 (N1) generates a low stable steady-state of cyclin B (P1), while r = 2.5 (N4) a high stable steady-state of cyclin B (P4). These stable steady-states are indicated by the intersections of the nullclines (filled circles). The unstable steady states are labeled with open circles (P2 and P3). (**D**) Relationship between the oscillation baseline and amplitude values and ATP concentration (positively correlated with r). Error bars indicate the ranges of 3 replicates. Inserts show two example time courses of total cyclin B with different r values (L1, r = 0.8; L2, r = 1.5), colors of which match the ones in **Figure 3C**. Simulation is done using Gillespie algorithm. (**E**) Time series of total cyclin B molecules from the model without ATP (top panel, green line) and with ATP (bottom panel, yellow line).

DOI: https://doi.org/10.7554/eLife.33549.009

*Figure 3 continued on next page*

*Figure 3 continued*

The following source data and figure supplement are available for figure 3:

**Source data 1.** Simulated data for generating *Figure 3—figure supplement 1* using the energy depletion model.
DOI: https://doi.org/10.7554/eLife.33549.011

**Figure supplement 1.** Analysis of the dependence of oscillations on the cyclin B synthesis and degradation rates.
DOI: https://doi.org/10.7554/eLife.33549.010

phosphorylation of Cdc25 and Wee1 can decrease R by activating Cdc25 and inhibiting Wee1 simultaneously, resulting in a high dependence of R on the ATP concentration (*Figure 3B*).

Using this model, we further investigated the relationship between ATP and the oscillation behaviors. We introduce a parameter r into our system to systematically change the ratio R (see Materials and methods 8). In *Figure 3C*, the phase plot of the two-ODE model shows that at a low r (e.g. 0.5), the system stays at a stable steady-state with low cyclin B concentration and at a high r (e.g. 2.5), the oscillation is arrested at a stable steady-state with high cyclin B concentration. At an intermediate value, increasing r can produce oscillations of increased amplitude, baseline and period (*Figure 3C,D*). If we assume that the available ATP concentration decreases over time, we can readily recapitulate the experimentally observed increment of amplitude, baseline, and period of the cyclin B time course (*Figure 3E*). The energy depletion model can also predict the experimental observations in *Figure 2D and E* by showing that both period and number of cycles decrease with increasing cyclin B concentration (*Figure 3—figure supplement 1C*, *Figure 3—source data 1*).

We noted that, besides phosphorylation, other processes, including protein synthesis and ubiquitination-mediated degradation, also consume ATPs and are sensitive to the energy level. However, the changes of synthesis and degradation rates yielded no obvious effects on the amplitude and baseline (*Figure 3—figure supplement 1D*).

We have developed here a novel artificial cell system that enables highly robust and tunable mitotic oscillations. The system is amenable to high throughput, quantitative manipulation and analysis of both cytoplasmic and nuclear processes. Given cell cycles share common topologies with many biological oscillators, the system may be valuable to investigate fundamental principles of oscillator theory.

Our energy depletion model suggested an interesting mechanism to modulate oscillations with a single control parameter r that depends on the energy-tunable balance of two positive feedback loops. Considering that the rapid, synchronous cleavages of an early embryo require a large amount of energy that remains unchanged for the first cleavage stages before rapidly dropping until the mid-blastula stage when cell cycles slow down (*Zotin et al., 1967*), this energy-dependent control may function as a 'checkpoint' to regulate cell cycles when r becomes large.

## Materials and methods

**Key resources table**

| Reagent type (species) or resource | Designation | Source or reference | Identifiers | Additional information |
|---|---|---|---|---|
| Biological sample (*Xenopus laevis*) | *Xenopus* eggs | Xenopus-I Inc. | RRID:NXR_0.0080 | |
| recombinant DNA reagent | securin-mCherry | this paper | | Progentiors: PCR; Gateway vector pMTB2 |
| recombinant DNA reagent | cyclin B1-YFP | this paper | | |
| recombinant DNA reagent | GST-GFP-NLS | PMID: 23863935 | | Dr. James Ferrell's lab |
| peptide, recombinant protein | GFP-NLS protein | this paper | | BL21 (DE3)-T-1 competent cells |
| commercial assay or kit | QIAprep spin miniprep kit | QIAGEN | Cat No.: 27104 | |
| commercial assay or kit | mMESSAGE mMACHINE SP6 Transcription Kit | Ambion | Cat No.: AM1340 | |
| Strain, strain background | BL21 (DE3)-T-1 competent celss | Sigma-Aldrich | Cat No.: B2935 | |

*Continued on next page*

*Continued*

| Reagent type (species) or resource | Designation | Source or reference | Identifiers | Additional information |
|---|---|---|---|---|
| chemical compound, drug | calcium ionophore | Sigma-Aldrich | Cat No.: A23187 | |
| chemical compound, drug | Hoechst 33342 | Sigma-Aldrich | Cat No.: B2261 | |
| chemical compound, drug | Trichloro (1H,1H,2H,2H-perfluorooctyl) silane | Sigma-Aldrich | Cat No.: 448931 | |
| chemical compound, drug | PFPE-PEG surfactant | Ran Biotechnologies | 008-FluoroSurfactant-2wtH-50G | |
| other | GE Healthcare Glutathione Sepharose 4B beads | Sigma-Aldrich | Cat No.: GE17-0756-01 | |
| other | PD-10 column | Sigma-Aldrich | Cat No.: GE17-0851-01 | |
| other | VitroCom miniature hollow glass tubing | VitroCom | Cat No.: 5012 | |

## Cycling *Xenopus laevis* extract preparation

Cycling *Xenopus* extracts were prepared as described (*Murray, 1991*), except that eggs were activated with calcium ionophore A23187 (200 ng/µL) rather than electric shock. Freshly prepared extracts were kept on ice while applied with de-membranated sperm chromatin (to approximately 250 per µl of extract), GFP-NLS (10 µM) and recombinant mRNAs of securin-mCherry (10 ng/µL) and cyclin B1-YFP (ranging from 0 to 10 ng/µL). The extracts were mixed with surfactant oil 2% PFPE-PEG to generate droplets.

## Fluorescence-labeled reporters

GFP-NLS protein was expressed in BL21 (DE3)-T-1 competent cells (Sigma Aldrich, B2935) that were induced by 0.1 mM IPTG (Isopropyl β-D-1-thiogalactopyranoside, Sigma Aldrich, I6758) overnight. Cells were broken down to release protein through sonication. GE Healthcare Glutathione Sepharose 4B beads (Sigma Aldrich, GE17-0756-01) and PD-10 column (Sigma Aldrich, GE17-0851-01) were used to purify and elute GFP-NLS protein. 200 mg/ml Hoechst 33342 (Sigma Aldrich, B2261) was added to stain chromosomes. Securin-mCherry and cyclin B1-YFP plasmids were constructed using Gibson assembly method (*Gibson et al., 2009*). All mRNAs were transcribed in vitro and purified using mMESSAGE mMACHINE SP6 Transcription Kit (Ambion, AM1340).

## Teflon-coated microchamber preparation

VitroCom miniature hollow glass tubing with height of 100 µm (VitroCom, 5012) was cut into pieces with lengths of 3–5 mm. A heating block was heated up to 95°C in a Fisher Scientific Isotemp digital incubator and then it was placed into a Bel-art F42025-0000 polycarbonate vacuum desiccator with white polypropylene bottom. The cut glass tubes and a 1.5 ml Eppendorf tube containing 30 µl Trichloro (1 hr,1H,2H,2H-perfluorooctyl) silane (Sigma Aldrich, 448931) were placed in the heating block. Vacuum was applied to the desiccator and the tubes was left incubated overnight.

## Generation of droplet-based artificial cells

To generate droplets, we used a Fisher Scientific vortex mixer to mix 20 µl cycling extract reaction mix, and 200 µl 2% PFPE-PEG surfactant (Ran Biotechnologies, 008-FluoroSurfactant-2wtH-50G) at speed level 10 for 3 s. By adjusting the vibration speed and ratio between aqueous and oil phase appropriately, we can obtain droplets with various sizes, ranging from 10 µm to 300 µm.

## Time-lapse fluorescence microscopy

All imaging was conducted on an Olympus FV1200 confocal microscope under MATL mode (multiple area time lapse) and Olympus IX83 microscope equipped with a motorized x-y stage, at room temperature. Time-lapse images were recorded in bright-field and multiple fluorescence channels at a time interval of 6–9 min for at least 12 hr up to four days.

## Image analysis and data processing

We used Imaris 8.1.2 (Bitplane Inc.) for image processing. Level-set method on bright-field images was used for droplet segmentation, and autoregressive motion algorithm was used for tracking. Tracks that had less than two oscillations were discarded. Results were then manually curated for accuracy. Means and standard deviations of fluorescence intensities as well as areas of each droplet were calculated for further analysis. The volume of a droplet was calculated using the formula proposed by a previous study (*Good et al., 2013*). To compensate for intensity drift over time, fluorescence intensity in droplets were normalized by average intensity of the background. For period calculation, Matlab (Mathworks Inc.) was used to detect peaks and troughs over the signal of mean intensity for cyclin B-YFP and securin-mCherry. All peaks were manually curated and edited to ensure reliability.

## Statistical analysis

In *Figure 2D–E*, error bars indicate the median absolute deviations (MAD) of measurements pooled across multiple loading replicates. Each loading replicate refers to droplets encapsulated with extracts prepared from the same batch of *Xenopus* eggs that were loaded into a distinct individual Teflon-coated micro-chamber. For cyclin B mRNA concentrations of 0, 3, 5, 8, 10 ng/µL, respectively two, two, one, two and three loading replicates were performed, resulting in sample sizes of 373, 443, 227, 430 and 554 droplets that were analyzed. Extracts with cyclin B mRNA concentrations above 10 ng/µL did not exhibit sustained oscillations before arresting at a mitotic phase.

In *Figure 2F–G*, droplets generated with three cyclin B mRNA concentrations (0, 3, 5 ng/µL) were analyzed, with samples sizes of 246, 475, and 177 respectively. Experiments with cyclin B mRNA concentrations higher than 5 ng/µL showed similar behaviors as that of the 5 ng/µL cyclin B mRNA experiment and therefore are not included in the figures. Colored areas are defined by first and third quantiles after LOWESS (Locally Weighted Scatterplot Smoothing) with a window size of 20. The quartiles are calculated using running percentile with a binning size of 50. Volumes of droplets were first estimated using the compressed diameters (*Good et al., 2013*). The equivalent diameter was then calculated by the diameter of a sphere with the same volume as the estimated volume of a droplet.

In *Figure 3B and D*, we performed simulations at chosen ranges of ATP percentage (*Figure 3B*) and r (*Figure 3D*) to yield self-sustained oscillations. Error bars indicate the ranges of 3 independent simulations. The time courses in the figure inserts are randomly selected from the simulation replicates as examples.

In *Figure 3—figure supplement 1A and B*, measurements from experiments with 0 ng/µL cyclin B mRNA (from two loading replicates, 373 droplets, 5820 cycles) were pooled to analyze the change of period over time. In *Figure 3—figure supplements 1C*, 50 independent simulations were performed for each cyclin B mRNA concentration. Error bars indicate the standard deviations of the simulated data. In *Figure 3—figure supplement 1D*, the ranges of r and the degradation/synthesis rate were chosen to ensure all points in grids have self-sustained oscillations. All simulations in the supplementary figures were performed in a long enough time window to ensure stable statistics (period, amplitude) while the ATP concentration remained positive.

## A two-ODE model of the embryonic cell cycle and stochastic simulations

Complicated models have been proposed to describe the embryonic cell cycle oscillation (*Novak and Tyson, 1993a*; *Ciliberto et al., 2003*; *Pomerening et al., 2005*; *Tsai et al., 2008*). However, simple two-ODE models with fewer parameters are more amenable to analysis, while still capturing the general property of the feedback loops. We described the net productions of cyclin B1 and active cyclinB-Cdk1 complex $[Cdk1_a]$ using the following two equations (*Yang and Ferrell, 2013*; *Tsai et al., 2014*):

$$\frac{d}{dt}[CyclinB] = k_{sy} - k_{\text{deg}}[CyclinB] = k_{sy} - \left( a_{deg} + \frac{b_{deg}[Cdk1_a]^{n_{deg}}}{[Cdk1_a]^{n_{deg}} + EC50_{deg}^{n_{deg}}} \right)[CyclinB] \qquad (1)$$

$$\frac{d}{dt}[Cdk1_a] = k_{sy} + k_{\text{Cdc25}}[Cdc25 - Pi]([CyclinB] - [Cdk1_a]) - k_{\text{Wee1}}[Wee1][Cdk1_a]$$
$$- k_{deg}[Cdk1_a]$$
$$= k_{sy}$$
$$+ \frac{1}{\sqrt{r}}\left(a_{Cdc25} + \frac{b_{Cdc25}[Cdk1_a]^{n_{Cdc25}}}{[Cdk1_a]^{n_{Cdc25}} + EC50_{Cdc25}^{n_{Cdc25}}}\right)([CyclinB] - [Cdk1_a])$$
$$- \sqrt{r}\left(a_{Wee1} + \frac{b_{Wee1}EC50_{Wee1}^{n_{Wee1}}}{[Cdk1_a]^{n_{Wee1}} + EC50_{Wee1}^{n_{Wee1}}}\right)[Cdk1_a]$$
$$- \left(a_{deg} + \frac{b_{deg}[Cdk1_a]^{n_{deg}}}{[Cdk1_a]^{n_{deg}} + EC50_{deg}^{n_{deg}}}\right)[Cdk1_a]$$

(2)

The parameters for the model are listed below:

| $k_{sy}$ | 1 nM/min |
|---|---|
| $a_{wee1}$ | 0.08 nM/min |
| $b_{wee1}$ | 0.4 nM/min |
| $n_{wee1}$ | 3.5 |
| $EC50_{wee1}$ | 35 nM |
| $a_{cdc25}$ | 0.16 nM/min |
| $b_{cdc25}$ | 0.8 nM/min |
| $n_{cdc25}$ | 11 |
| $EC50_{cdc25}$ | 30 nM |
| $a_{deg}$ | 0.01 nM/min |
| $b_{deg}$ | 0.04 nM/min |
| $n_{deg}$ | 17 |
| $EC50_{deg}$ | 32 nM |

Here, $[CyclinB]$ and $[Cdk1_a]$ refer to the concentrations of cyclin B1 and active cyclin B1-Cdk1 complex. $[Wee1]$ is the concentration of active Wee1, while $[Cdc25 - Pi]$ is the concentration of active Cdc25. We assumed that Cyclin B1 is synthesized at a constant rate. Its degradation rate is dependent on Cdk1 activity in the form of a Hill function with an exponent of 17 (*Yang and Ferrell, 2013*). Active cyclin B1-Cdk1 complex can also be eliminated through cyclin degradation. In addition, we considered the concentration of Cdk1 to be high compared to the peak concentration of cyclin B1 (*Hochegger et al., 2001*; *Kobayashi et al., 1991*) and the affinity of these cyclins for Cdk1 to be high (*Kobayashi et al., 1994*). Thus, there is no unbound form of cyclin B1, and the newly synthesized cyclin B1 is converted to cyclin-Cdk1 complexes, which are rapidly phosphorylated by the Cdk-activating kinase CAK to produce active Cdk1. According to previous studies, these complexes can then be inactivated by Wee1 and re-activated by Cdc25, via the double-negative and positive

**Table 1.** Reaction rates and stoichiometry of the two-ODE model.

| Reaction | Rate | Stoichiometry |
|---|---|---|
| Active Cdk1 Synthesis | $\rho_1 = k_{sy}$ | $<Cdk1_a> = <Cdk1_a> + 1$ |
| Active Cdk1 to Inactive Cdk1 | $\rho_2 = \sqrt{r}\left(a_{Wee1} + \frac{b_{Wee1}EC50_{Wee1}^{n_{Wee1}}}{<Cdk1_a>^{n_{Wee1}} + EC50_{Wee1}^{n_{Wee1}}}\right)<Cdk1_a>$ | $<Cdk1_a> = <Cdk1_a> - 1 \; <Cdk1_i> = <Cdk1_i> + 1$ |
| Inactive Cdk1 to Active Cdk1 | $\rho_3 = \frac{1}{\sqrt{r}}\left(a_{Cdc25} + \frac{b_{Cdc25}<Cdk1_a>^{n_{Cdc25}}}{<Cdk1_a>^{n_{Cdc25}} + EC50_{Cdc25}^{n_{Cdc25}}}\right)<Cdk1_i>$ | $<Cdk1_a> = <Cdk1_a> + 1 \; <Cdk1_i> = <Cdk1_i> - 1$ |
| Active Cdk1 Degradation | $\rho_4 = \left(a_{deg} + \frac{b_{deg}<Cdk1_a>^{n_{deg}}}{<Cdk1_a>^{n_{deg}} + EC50_{deg}^{n_{deg}}}\right)<Cdk1_a>$ | $<Cdk1_a> = <Cdk1_a> - 1$ |
| Inactive Cdk1 Degradation | $\rho_5 = \left(a_{deg} + \frac{b_{deg}<Cdk1_a>^{n_{deg}}}{<Cdk1_a>^{n_{deg}} + EC50_{deg}^{n_{deg}}}\right)<Cdk1_i>$ | $<Cdk1_i> = <Cdk1_i> - 1$ |

DOI: https://doi.org/10.7554/eLife.33549.012

**Table 2.** Reaction rates in the model considering ATP.

| Reaction | Rate |
|---|---|
| Active Cdk1 Synthesis | $\rho_1 = k_{sy}$ |
| Active Cdk1 to Inactive Cdk1 | $\rho_2 = 2[ATP]<Cdk1_a> \left( a_{Wee1} + \frac{b_{Wee1}EC50_{Wee1}^{n_{Wee1}}}{<Cdk1_a>^{n_{Wee1}}+EC50_{Wee1}^{n_{Wee1}}} \right) \left( \frac{1-[ATP]}{[ATP]\left(1-\frac{2[Wee1_0]}{[Wee1_{tot}]}\right)+\frac{[Wee1_0]}{[Wee1_{tot}]}} \right)$ |
| Inactive Cdk1 to Active Cdk1 | $\rho_3 = <Cdk1_i> \left( a_{Cdc25} + \frac{b_{Cdc25}<Cdk1_a>^{n_{Cdc25}}}{<Cdk1_a>^{n_{Cdc25}}+EC50_{Cdc25}^{n_{Cdc25}}} \right) \left( \frac{[ATP]}{1-\frac{[Cdc25-Pi_0]}{[Cdc25_{tot}]}+\left(2^{\frac{[Cdc25-Pi_0]}{[Cdc25_{tot}]}}-1\right)[ATP]} \right)$ |
| Active Cdk1 Degradation | $\rho_4 = \left( a_{deg} + \frac{b_{deg}<Cdk1_a>^{n_{deg}}}{<Cdk1_a>^{n_{deg}}+EC50_{deg}^{n_{deg}}} \right) <Cdk1_a>$ |
| Inactive Cdk1 Degradation | $\rho_5 = \left( a_{deg} + \frac{b_{deg}<Cdk1_a>^{n_{deg}}}{<Cdk1_a>^{n_{deg}}+EC50_{deg}^{n_{deg}}} \right) <Cdk1_i>$ |

DOI: https://doi.org/10.7554/eLife.33549.013

feedback loops, with Hill exponent of $n_{Wee1}$ as 3.5 and $n_{Cdc25}$ as 11 (**Kim and Ferrell, 2007**; **Trunnell et al., 2011**).

We use a free parameter $r$, representing the ratio of the double negative and double positive feedback strengths, to permute the balance between the two feedbacks. This balance is suggested to be critical for oscillatory properties (**Tsai et al., 2014**). Note that this r is a parameter while R in the main text is a measurement that changes over a simulation.

In droplets that have small volumes and contain small numbers of molecules, the stochastic nature of the underlying biochemical reactions must be considered. We adapted a stochastic two-ODE model (**Yang and Ferrell, 2013**), and converted our two-ODE model to the corresponding chemical master equations (**Kampen, 1992**) and carried out numerical simulations using the Gillespie algorithm (**Gillespie, 1977**). The reaction rates and molecular stoichiometry are shown in **Table 1**.

## A stochastic model of the embryonic cell cycle including energy effect

To explore how energy consumption could affect the oscillations, we took ATP into account for phosphorylation and dephosphorylation of Wee1 (**Tuck et al., 2013**), such that:

$$Wee1+ATP \rightleftharpoons Wee1-Pi+ADP \qquad (3)$$

In our model, we assumed Wee1 is in equilibrium with the activity of Cdk1 due to fast reactions between Cdk1 and Wee1. Using the reaction coefficients for Wee1 phosphorylation as $k_{1Wee1}$ and that for Wee1-Pi dephosphorylation as $k_{2Wee1}$, along with the steady-state approximation, we have:

$$k_{1Wee1}[Wee1][ATP] = k_{2Wee1}[Wee1-Pi][ADP] = k_{2Wee1}([Wee1_{tot}]-[Wee1])(1-[ATP]) \qquad (4)$$

All above modifications for Wee1 reactions were also applied to Cdc25. After normalizing [ATP] and [ADP] by $[ATP]+[ADP]$, we have the updated reaction rates summarized in **Table 2**. Here the $[wee1]_0$ and $[cdc25\text{-}Pi]_0$ represent the steady-state concentration of active Wee1 and Cdc25 when ATP is not considered in reaction. The ratios of the steady-state to total concentrations of Wee1 and Cdc25 can be calculated as a function of active Cdk1 using the parameters from previous work (**Novak and Tyson, 1993**).

## Acknowledgements

We thank Madeleine Lu for constructing securin-mCherry plasmid, Lap Man Lee and Kenneth Ho for discussions about droplet generation, Neha Bidthanapally and Zheng Yang for helping image processing, Jeremy B. Chang and James E. Ferrell Jr for providing GFP-NLS construct. QY is funded by the National Science Foundation (Early CAREER Grant #1553031), the National Institutes of Health (MIRA #GM119688), and a Sloan Research Fellowship. AP is funded by the National Science Foundation (MCB #1612917).

## Additional information

### Funding

| Funder | Grant reference number | Author |
|---|---|---|
| National Science Foundation | MCB #1612917 | Allen P Liu |
| National Institutes of Health | MIRA #GM119688 | Qiong Yang |
| National Science Foundation | CAREER Grant #1553031 | Qiong Yang |
| Alfred P. Sloan Foundation | Sloan Research Fellowship | Qiong Yang |

The funders had no role in study design, data collection and interpretation, or the decision to submit the work for publication.

### Author contributions

Ye Guan, Data curation, Software, Formal analysis, Validation, Investigation, Visualization, Methodology, Writing—original draft, Writing—review and editing; Zhengda Li, Software, Formal analysis, Visualization, Writing—original draft, Writing—review and editing; Shiyuan Wang, Formal analysis, Investigation; Patrick M Barnes, Xuwen Liu, Software, Formal analysis; Haotian Xu, Software, Help with data analysis; Minjun Jin, Resources, Investigation; Allen P Liu, Resources, Writing—review and editing; Qiong Yang, Conceptualization, Software, Formal analysis, Supervision, Funding acquisition, Visualization, Methodology, Writing—original draft, Project administration, Writing—review and editing

### Author ORCIDs

Allen P Liu (iD) http://orcid.org/0000-0002-0309-7018
Qiong Yang (iD) http://orcid.org/0000-0002-2442-2094

### Decision letter and Author response

Decision letter https://doi.org/10.7554/eLife.33549.029
Author response https://doi.org/10.7554/eLife.33549.030

## Additional files

### Supplementary files

• Source code 1. Source code to generate *Figure 1D* based on raw image file.
DOI: https://doi.org/10.7554/eLife.33549.014

• Source code 2. Source code to generate *Figure 2D–E* and *Figure 2F–G* using *Figure 2—source data 1* and *Figure 2—source data 2* respectively.
DOI: https://doi.org/10.7554/eLife.33549.015

• Source code 3. Source code to generate *Figure 3B–E*.
DOI: https://doi.org/10.7554/eLife.33549.016

• Source code 4. Source code to generate *Figure 3—figure supplement 1A–B* using *Figure 3—source data 1*.
DOI: https://doi.org/10.7554/eLife.33549.017

• Transparent reporting form
DOI: https://doi.org/10.7554/eLife.33549.018

### Major datasets

The following dataset was generated:

| Author(s) | Year | Dataset title | Dataset URL | Database, license, and accessibility information |
|---|---|---|---|---|
| Ye Guan, Zhengda Li, Shiyuan Wang, Patrick M Barnes, Xuwen Liu, Haotian Xu, Minjun Jin, Qiong Yang, Allen P Liu | 2018 | Data from: A robust and tunable mitotic oscillator in artificial cells | http://dx.doi.org/10. 5061/dryad.2929bs2 | Available at Dryad Digital Repository under a CC0 Public Domain Dedication. |

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
