## [Decision Letter]

Thank you for submitting your article "A robust and tunable mitotic oscillator in artificial cells" for consideration by *eLife*. Your article has been reviewed by two peer reviewers, and the evaluation has been overseen by Naama Barkai as the Senior and Reviewing Editor. The reviewers have opted to remain anonymous.

The reviewers have discussed the reviews with one another and the Reviewing Editor has drafted this decision to help you prepare a revised submission. Please address in full all comments of reviewer #2.

*Reviewer #1:*

Guan et al. present a highly interesting manuscript showing that a few simple components in the cell cycle machinery can generate oscillations in vitro in droplets. They showed robust oscillatory behaviors and how the oscillations can be tuned by the size of the droplets as well as by concentrations of cyclin mRNAs. The theoretical work showed that the ratio of Cdc25 and Wee1 affect the oscillation timing. They further showed that the energy depletion model is consistent with the experimental results. The experiments are elegantly carried out and the results are beautiful. I would recommend for publication in *eLife*.

*Reviewer #2:*

This paper reports on the development of a droplet assay in which cytoplasmic extracts can be compartmentalized to sizes in the micrometer range. This allows the authors to generate artificial mitotic oscillations using an in vitro setup that combines the ease of manipulation and control of bulk cell-free extracts with the realism of cell-size compartmentalization (in terms of heterogeneity and sustained dynamics). The observed sustained oscillations (Figure 2B) are certainly noteworthy, and the assay does allow for a valuable study of how the period of the oscillator depends on "cell" size and on the availability of one of the clock components (cyclin B1). The setup also enables the authors to conjecture that the amplitude and period of the oscillator depend on the energy available to the cell in terms of ATP, a fact that is studied computationally but not experimentally. I would like to raise the following points about this work:

1) The oscillations shown in Figure 1C, D are really interesting, given that they exist even in the presence of nuclear self-assembly and disassembly. I was wondering if the extra compartmentalization provided by the nucleus would have an impact on the oscillations reported in Figure 2. Would the authors be able to obtain self-sustained oscillations for long periods in the presence of a nucleus? This would bring the results closer to real-life situations.

2) Figures 2F and G show that the period and lifetime of the oscillations are rather independent of "cell" sizes for large sizes. I think it'd be interesting to relate these results to real cells, by taking into account the range of sizes that real cells exhibit (keeping in mind again that these "cells" don't have a nucleus, see my preceding comment).

3) The way the modeling results are being presented makes it difficult to understand what is being done. The title of the caption of Figure 3 refers to a "stochastic model", but as far as I understand the results of Figure 3C have been obtained with an ODE model, in the absence of fluctuations, and probably those of Figure 3E too. Figure 3B, on the other hand, has been presumably obtained with a stochastic model (since this seems to be the way in which ATP can be included in the model described in the Materials and methods section), before the stochastic description is mentioned in the text. On the other hand, the time traces in the insets look very clean, seemingly obtained from an ODE model. A similar comment can be made regarding Figure 3D. Can the authors clarify what is being shown in Figure 3?

4) Figure 3C is really hard to follow. I would ask the authors to plot the steady states (at least the stable ones, and preferably all) as symbols, and to represent the trajectories in colors different from those of the corresponding nullclines.

Finally, I have two suggestions that are not essential to support the major conclusions of the paper, but would (in my opinion) enrich it:

5) I'm wondering if the range of droplet sizes that has been studied has been selected on purpose by the authors, or if there is some limitation resulting from the experimental process. I'm asking because it would be interesting, in my opinion, to see what happens with the oscillations as the size of the droplets increases. For what droplet size does the oscillation dampening typical of bulk extracts is recovered in these experiments?

6) It would also be desirable to validate the modeling results of Figure 3 experimentally by controlling experimentally the amount of available ATP and measure the corresponding effect on the oscillation characteristics.

---

## [Author Response]

Reviewer #1:Guan et al. present a highly interesting manuscript showing that a few simple components in the cell cycle machinery can generate oscillations in vitro in droplets. They showed robust oscillatory behaviors and how the oscillations can be tuned by the size of the droplets as well as by concentrations of cyclin mRNAs. The theoretical work showed that the ratio of Cdc25 and Wee1 affect the oscillation timing. They further showed that the energy depletion model is consistent with the experimental results. The experiments are elegantly carried out and the results are beautiful. I would recommend for publication in eLife.

We thank reviewer 1 for this positive assessment of our work. We are excited that you enjoyed our work.

Reviewer #2:This paper reports on the development of a droplet assay in which cytoplasmic extracts can be compartmentalized to sizes in the micrometer range. This allows the authors to generate artificial mitotic oscillations using an in vitro setup that combines the ease of manipulation and control of bulk cell-free extracts with the realism of cell-size compartmentalization (in terms of heterogeneity and sustained dynamics). The observed sustained oscillations (Figure 2B) are certainly noteworthy, and the assay does allow for a valuable study of how the period of the oscillator depends on "cell" size and on the availability of one of the clock components (cyclin B1). The setup also enables the authors to conjecture that the amplitude and period of the oscillator depend on the energy available to the cell in terms of ATP, a fact that is studied computationally but not experimentally. I would like to raise the following points about this work:1) The oscillations shown in Figure 1C, D are really interesting, given that they exist even in the presence of nuclear self-assembly and disassembly. I was wondering if the extra compartmentalization provided by the nucleus would have an impact on the oscillations reported in Figure 2. Would the authors be able to obtain self-sustained oscillations for long periods in the presence of a nucleus? This would bring the results closer to real-life situations.

To investigate the effect of the nucleus, we compared the oscillation behaviors of two groups of droplets prepared from the same batch of extracts in the same day, one group with demembranated *Xenopus* sperm chromatin added and one without. The presence of nuclear self-assembly and disassembly seems to reduce the overall lifespan of oscillations (Figure 1C) and lengthen the periods (Figure 1B). The reduced lifespan of oscillations could be explained by faster energy depletion in droplets with nuclei, as nuclear pore transportation, chromosome condensation and the nuclear envelope formation/breakdown would consume energy. Moreover, compared to the cytoplasmic-only droplets, cyclin-CDKs in droplets with nuclei would need to phosphorylate more substrates required for regulating the downstream nuclear events and the transportation of molecules into/out of nucleus would take time, which could explain why adding sperm DNA may elongate the cycle period. Together, the reduced lifespan and the extended oscillation periods lead to a smaller number of oscillations (Figure 1A).

We also observed that the effect of adding sperm DNAs is more obvious in large droplets (Figure 1D, E), likely due to the fact that some small droplets may have no sperm DNA encapsulated.

**Author response image 1. respfig1:** A-C Statistics of droplets with sperm DNA (N=320) and without sperm DNA (N=373). All droplets were prepared in the same day from the same batch of frog eggs. The figures show that adding sperm DNA may cause significant difference in number of oscillations (**A**), period (**B**) and oscillation sustaining time (**C**). D-E. Droplet-size dependence of period (**D**) and number of oscillations (**E**) with nucleus effect, showing that adding sperm DNAs has a more significant impact on larger droplets.

2) Figures 2F and G show that the period and lifetime of the oscillations are rather independent of "cell" sizes for large sizes. I think it'd be interesting to relate these results to real cells, by taking into account the range of sizes that real cells exhibit (keeping in mind again that these "cells" don't have a nucleus, see my preceding comment).

We thank reviewer 2 for bringing up this interesting question. The real cell size spans a large range. An egg cell like *Xenopus* or zebrafish is around 1mm in diameter. Upon fertilization, the embryo undergoes rapid and reductive divisions in the absence of growth, cleaving it into smaller and smaller cells down to 10 μm in diameter. A cultured mammalian cell has a size of around 20 µm.

In our system, droplet size varies from around 200 µm (large droplets) in diameter to around 10 µm (small droplets), covering the range of sizes from an embryonic cell after the 7^th^ cleavage to a somatic-sized blastomere. A direct comparison of our droplets to embryonic cells might not be feasible since these droplets do not have a nucleus. In addition, most of the embryo volume is taken by yolk and the relevant cytosol volume of the embryo might only take up a small portion. However, we could explore the mechanism of the size effect on the cell cycle period within our droplet system that may be extendable to real cells.

In our ongoing project, we are trying to relate the size dependency of period to that observed in actual embryo cleavages. Published results on early embryo cleavages in both zebrafish (Citation: Figure 4A and 4B in Olivier et al., Science 2010) and *Xenopus* (Citation: Figure 2B and 2F in Anderson et al., Cell Report 2017) support that the cell cycle accelerates at the beginning of the cleavages and slows down as cells get smaller (Anderson et al., 2017; Olivier et al., 2010). The same result could be observed in our system if we assume a large droplet of 200 µm in diameter undergoes multiple cycles of cleavages in a thought experiment (Author response image 2). Here we sorted the droplets in Figure 2F of the main text based on their size to simulate the first ten cleavages. The periods of all droplets within certain diameters are grouped. The result suggests that when droplets are getting smaller, their periods decrease a little at the beginning and then increase, comparable to the trend observed in early embryonic divisions.

**Author response image 2. respfig2:** A thought experiment of a droplet of 200 µm in diameter undergoing ten divisions. The cell cycle period is calculated for each division, with each corresponding diameter labeled on the figure. The error bar is showing 25^th^ and 75^th^ percentile as in Figure 2F in the main text.

3) The way the modeling results are being presented makes it difficult to understand what is being done. The title of the caption of Figure 3 refers to a "stochastic model", but as far as I understand the results of Figure 3C have been obtained with an ODE model, in the absence of fluctuations, and probably those of Figure 3E too. Figure 3B, on the other hand, has been presumably obtained with a stochastic model (since this seems to be the way in which ATP can be included in the model described in the Materials and methods section), before the stochastic description is mentioned in the text. On the other hand, the time traces in the insets look very clean, seemingly obtained from an ODE model. A similar comment can be made regarding Figure 3D. Can the authors clarify what is being shown in Figure 3?

We thank reviewer 2 for the careful thought and for raising these points. To clarify, Figure 3C and Figure 3E top panel from the main text are not stochastic and are obtained from an ODE model; others (Figures 3B, 3D, and 3E bottom panel of the main text) are from a stochastic model. The simulation code has been included in the source manuscript files. To avoid the confusion, we have removed “stochastic model” from the title of Figure 3 in the main text. We have also modified Figure 3C and Figure 3D for clearer presentation of the modeling results.

The stochastically simulated trace looks smooth because we assumed a system of 1 pL in volume (comparable to our typical droplet volume) and the total number of molecules is high enough to produce a deterministic ODE-like response. As an example, we replot Figure 3E bottom panel from the main text to show the effects of the volume (or the total number of molecules). As shown in Author response image 3, when droplet size is above 0.1 pL, no obvious kinks can be observed. Note that we have used the standard Gillespie algorithm for stochastic simulation, which considers the reaction stochasticity alone. But in real cases, the diffusive stochasticity might play a role, which is not considered in current simulation.

**Author response image 3. respfig3:** Reproduction of Figure 3E bottom panel from the main text with different reaction volumes.

4) Figure 3C is really hard to follow. I would ask the authors to plot the steady states (at least the stable ones, and preferably all) as symbols, and to represent the trajectories in colors different from those of the corresponding nullclines.

Figure 3C and the caption have been changed as reviewer 2 suggested.

Finally, I have two suggestions that are not essential to support the major conclusions of the paper, but would (in my opinion) enrich it:5) I'm wondering if the range of droplet sizes that has been studied has been selected on purpose by the authors, or if there is some limitation resulting from the experimental process. I'm asking because it would be interesting, in my opinion, to see what happens with the oscillations as the size of the droplets increases. For what droplet size does the oscillation dampening typical of bulk extracts is recovered in these experiments?

We thank reviewer 2 for this interesting question. We did not select the droplet sizes on purpose, but we performed all experiments at a fixed vortexing speed and duration. At this vortexing condition, we could obtain droplets of sizes ranging from about 10 µm to 300 µm (Author response image 4). Droplet sizes beyond 200 µm are rare. In an unusually large droplet (beyond 500 µm) such as the one example shown in Author response image 5, we could start to observe wave propagation across the droplet. Such spatial asynchrony (i.e. oscillations at different locations are out of phase) could potentially contribute to damped oscillations typical of a bulk ensemble measurement, even though oscillation at a specific location is sustained and has amplitude/period/lifetime comparable to a relatively smaller droplet. We could also take a population average of traces of all droplets in one experiment, to mimic the behaviors of bulk extracts.

**Author response image 4. respfig4:** Size distribution for 1997 droplets obtained using a specific vortexing speed and duration.

**Author response image 5. respfig5:** Mitotic trigger waves in a large droplet.

6) It would also be desirable to validate the modeling results of Figure 3 experimentally by controlling experimentally the amount of available ATP and measure the corresponding effect on the oscillation characteristics.

To explore the energy dependent behaviors experimentally, we tested droplets supplied with a variety of concentrations of energy mix. We define the standard concentration of energy mix, which we use in the article, as 1x. For all energy mix concentrations that we tested (ranging from 0 to 10x), we did not observe any oscillations in droplets with energy mix beyond 2x. In the following, we analyzed the oscillation behaviors in droplets prepared from the same batch of extracts except that they were supplied with different concentrations of energy mix (0x, 0.5x, 1x, 1.5x, with each condition performed in duplicate). As shown in Author response image 6, there is a clear trend of decreasing baseline, amplitude, and period as the amount of energy mix is increasing, which is consistent with our modeling results. The dependence of period and amplitude on the energy level is quantitatively analyzed in Author response image 7.

**Author response image 6. respfig6:** Time traces of individual droplets with different concentrations of energy mix. Each condition has two duplicate experiments and only one is shown. It is evident that the amplitude, baseline, and period decrease as the level of energy mix increases.

**Author response image 7. respfig7:** Amplitude (left panel) and period (right panel) decrease as energy mix concentration increases.